# Learning a Multi-View Stereo Machine

**Abhishek Kar**
UC Berkeley
akar@berkeley.edu

**Christian Häne**
UC Berkeley
chaene@berkeley.edu

**Jitendra Malik**
UC Berkeley
malik@berkeley.edu

## Abstract

We present a learnt system for multi-view stereopsis. In contrast to recent learning based methods for 3D reconstruction, we leverage the underlying 3D geometry of the problem through feature projection and unprojection along viewing rays. By formulating these operations in a differentiable manner, we are able to learn the system end-to-end for the task of metric 3D reconstruction. End-to-end learning allows us to jointly reason about shape priors while conforming to geometric constraints, enabling reconstruction from much fewer images (even a single image) than required by classical approaches as well as completion of unseen surfaces. We thoroughly evaluate our approach on the ShapeNet dataset and demonstrate the benefits over classical approaches and recent learning based methods.

## 1 Introduction

Multi-view stereopsis (MVS) is classically posed as the following problem - given a set of images with known camera poses, it produces a geometric representation of the underlying 3D world. This representation can be a set of disparity maps, a 3D volume in the form of voxel occupancies, signed distance fields etc. An early example of such a system is the stereo machine from Kanade *et al.* [26] that computes disparity maps from images streams from six video cameras. Modern approaches focus on acquiring the full 3D geometry in the form of volumetric representations or polygonal meshes [48]. The underlying principle behind MVS is simple - a 3D point looks locally similar when projected to different viewpoints [29]. Thus, classical methods use the basic principle of finding dense correspondences in images and triangulate to obtain a 3D reconstruction.

The question we try to address in this work is can we *learn* a multi-view stereo system? For the binocular case, Becker and Hinton [1] demonstrated that a neural network can learn to predict a depth map from random dot stereograms. A recent work [28] shows convincing results for binocular stereo by using an end-to-end learning approach with binocular geometry constraints.

In this work, we present Learnt Stereo Machines (LSM) - a system which is able to reconstruct object geometry as voxel occupancy grids or per-view depth maps from a small number of views, including just a single image. We design our system inspired by classical approaches while learning each component from data embedded in an end to end system. LSMs have built in projective geometry, enabling reasoning in metric 3D space and effectively exploiting the geometric structure of the MVS problem. Compared to classical approaches, which are designed to exploit a specific cue such as silhouettes or photo-consistency, our system learns to exploit the cues that are relevant to the particular instance while also using priors about shape to predict geometry for unseen regions.

Recent work from Choy *et al.* [5] (3D-R2N2) trains convolutional neural networks (CNNs) to predict object geometry given only images. While this work relied primarily on semantic cues for reconstruction, our formulation enables us to exploit strong geometric cues. In our experiments, we demonstrate that a straightforward way of incorporating camera poses for volumetric occupancy prediction does not lead to expected gains, while our geometrically grounded method is able to effectively utilize the additional information.

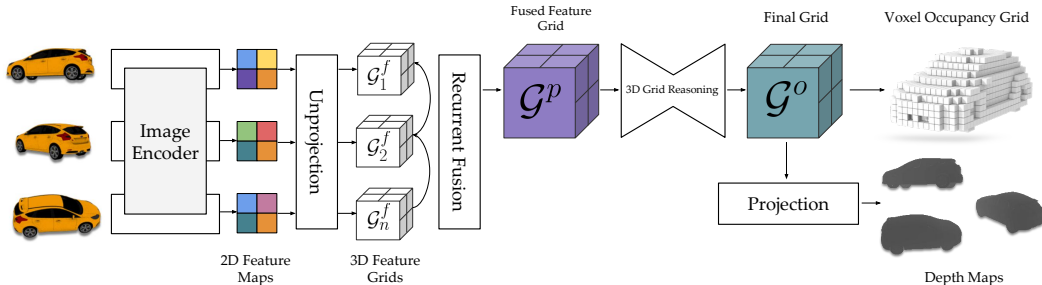

**Figure 1:** Overview of a Learnt Stereo Machine (LSM). It takes as input one or more views and camera poses. The images are processed through a feature encoder which are then unprojected into the 3D world frame using a differentiable unprojection operation. These grids $\{G_i^f\}_{i=1}^n$ are then matched in a recurrent manner to produce a fused grid $\mathcal{G}^p$ which is then transformed by a 3D CNN into $\mathcal{G}^o$. LSMs can produce two kinds of outputs - voxel occupancy grids (Voxel LSM) decoded from $\mathcal{G}^o$ or per-view depth maps (Depth LSM) decoded after a projection operation.

Classical multi-view stereopsis is traditionally able to handle both objects and scenes - we only showcase our system for the case of objects with scenes left for future work. We thoroughly evaluate our system on the synthetic ShapeNet [3] dataset. We compare to classical plane sweeping stereo, visual hulls and several challenging learning-based baselines. Our experiments show that we are able to reconstruct objects with fewer images than classical approaches. Compared to recent learning based reconstruction approaches, our system is able to better use camera pose information leading to significantly large improvements while adding more views. Finally, we show successful generalization to unseen object categories demonstrating that our network goes beyond semantic cues and strongly uses geometric information for unified single and multi-view 3D reconstruction.

## 2 Related Work

Extracting 3D information from images is one of the classical problems in computer vision. Early works focused on the problem of extracting a disparity map from a binocular image pair [36]. We refer the reader to [47] for an overview of classical binocular stereo matching algorithms. In the multi-view setting, early work focused on using silhouette information via visual hulls [32], incorporating photo-consistency to deal with concavities (photo hull) [29], and shape refinement using optimization [55, 50, 7, 15]. [39, 35, 54] directly reason about viewing rays in a voxel grid, while [34] recovers a quasi dense point cloud. In our work, we aim to learn a multi-view stereo machine grounded in geometry, that learns to use these classical constraints while also being able to reason about semantic shape cues from the data. Another approach to MVS involves representing the reconstruction as a collection of depth maps [6, 57, 41, 13, 40]. This allows recovery of fine details for which a consistent global estimate may be hard to obtain. These depth maps can then be fused using a variety of different techniques [38, 8, 33, 59, 30]. Our learnt system is able to produce a set of per-view depth maps along with a globally consistent volumetric representation which allows us to preserve fine details while conforming to global structure.

Learning has been used for multi-view reconstruction in the form of shape priors for objects [2, 9, 58, 20, 27, 52], or semantic class specific surface priors for scenes [22, 17, 45]. These works use learnt shape models and either directly fit them to input images or utilize them in a joint representation that fuses semantic and geometric information. Most recently, CNN based learning methods have been proposed for 3D reconstruction by learning image patch similarity functions [60, 18, 23] and end-to-end disparity regression from stereo pairs [37, 28]. Approaches which predict shape from a single image have been proposed in form of direct depth map regression [46, 31, 10], generating multiple depth maps from novel viewpoints [51], producing voxel occupancies [5, 16], geometry images [49] and point clouds [11]. [12] study a related problem of view interpolation, where a rough depth estimate is obtained within the system.

A line of recent works, complementary to ours, has proposed to incorporate ideas from multi-view geometry in a learning framework to train single view prediction systems [14, 56, 53, 42, 61] using multiple views as supervisory signal. These works use the classical cues of photo-consistency and

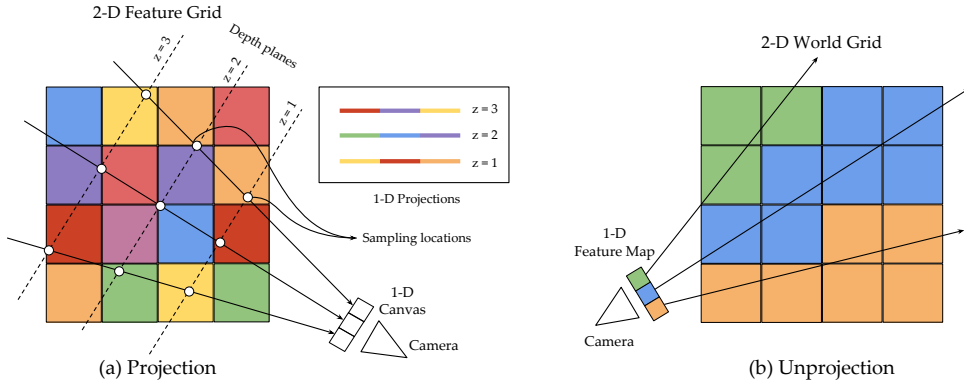

**Figure 2:** Illustrations of projection and unprojection operations between 1D maps and 2D grids. (a) The projection operation samples values along the ray at equally spaced z-values into a 1D canvas/image. The sampled features (shown by colors here) at the z planes are stacked into channels to form the projected feature map. (b) The unprojection operation takes features from a feature map (here in 1-D) and places them along rays at grid blocks where the respective rays intersect. Best viewed in color.

silhouette consistency only during training - their goal during inference is to only perform single image shape prediction. In contrast, we also use geometric constraints during inference to produce high quality outputs.

Closest to our work is the work of Kendall *et al.* [28] which demonstrates incorporating binocular stereo geometry into deep networks by formulating a cost volume in terms of disparities and regressing depth values using a differentiable arg-min operation. We generalize to multiple views by tracing rays through a discretized grid and handle variable number of views via incremental matching using recurrent units. We also propose a differentiable projection operation which aggregates features along viewing rays and learns a nonlinear combination function instead of using the differentiable arg-min which is susceptible to multiple modes. Moreover, we can also infer 3D from a single image during inference.

## 3 Learnt Stereo Machines

Our goal in this paper is to design an end-to-end learnable system that produces a 3D reconstruction given one or more input images and their corresponding camera poses. To this end, we draw inspiration from classical geometric approaches where the underlying guiding principle is the following - the reconstructed 3D surface has to be photo-consistent with all the input images that depict this particular surface. Such approaches typically operate by first computing dense features for correspondence matching in image space. These features are then assembled into a large cost volume of geometrically feasible matches based on the camera pose. Finally, the optimum of this matching volume (along with certain priors) results in an estimate of the 3D volume/surface/disparity maps of the underlying shape from which the images were produced.

Our proposed system, shown in Figure 1, largely follows the principles mentioned above. It uses a discrete grid as internal representation of the 3D world and operates in metric 3D space. The input images $\{I_i\}_{i=1}^n$ are first processed through a shared image encoder which produces dense feature maps $\{\mathcal{F}_i\}_{i=1}^n$, one for each image. The features are then *unprojected* into 3D feature grids $\{\mathcal{G}_i^f\}_{i=1}^n$ by rasterizing the viewing rays with the known camera poses $\{\mathcal{P}_i\}_{i=1}^n$. This unprojection operation aligns the features along epipolar lines, enabling efficient local matching. This matching is modelled using a recurrent neural network which processes the unprojected grids sequentially to produce a grid of local matching costs $\mathcal{G}^p$. This cost volume is typically noisy and is smoothed in an energy optimization framework with a data term and smoothness term. We model this step by a feed forward 3D convolution-deconvolution CNN that transforms $\mathcal{G}^p$ into a 3D grid $\mathcal{G}^o$ of smoothed costs taking context into account. Based on the desired output, we propose to either let the final grid be a volumetric occupancy map or a grid of features which is *projected* back into 2D feature

maps $\{\mathcal{O}_i\}_{i=1}^n$ using the given camera poses. These 2D maps are then mapped to a view specific representation of the shape such as a per view depth/disparity map. The key components of our system are the differentiable projection and unprojection operations which allow us to learn the system end-to-end while injecting the underlying 3D geometry in a metrically accurate manner. We refer to our system as a Learnt Stereo Machine (**LSM**). We present two variants - one that produces per voxel occupancy maps (**Voxel LSM**) and another that outputs a depth map per input image (**Depth LSM**) and provide details about the components and the rationale behind them below.

**2D Image Encoder.** The first step in a stereo algorithm is to compute a good set of features to match across images. Traditional stereo algorithms typically use raw patches as features. We model this as a feed forward CNN with a convolution-deconvolution architecture with skip connections (UNet) [44] to enable the features to have a large enough receptive field while at the same time having access to lower level features (using skip connections) whenever needed. Given images $\{I_i\}_{i=1}^n$, the feature encoder produces dense feature maps $\{\mathcal{F}_i\}_{i=1}^n$ in 2D image space, which are passed to the unprojection module along with the camera parameters to be lifted into metric 3D space.

**Differentiable Unprojection.** The goal of the unprojection operation is to lift information from 2D image frame to the 3D world frame. Given a 2D point $p$, its feature representation $\mathcal{F}(p)$ and our global 3D grid representation, we replicate $\mathcal{F}(p)$ along the viewing ray for $p$ into locations along the viewing ray in the metric 3D grid (a 2D illustration is presented in Figure 2). In the case of perspective projection specified by an intrinsic camera matrix $K$ and an extrinsic camera matrix $[R|t]$, the unprojection operation uses this camera pose to trace viewing rays in the world and copy the image features into voxels in this 3D world grid. Instead of analytically tracing rays, given the centers of blocks in our 3D grid $\{X_w^k\}_{k=1}^{N_V}$, we compute the feature for $k^{th}$ block by projecting $\{X_w^k\}$ using the camera projection equations $p'_k = K[R|t]X_w^k$ into the image space. $p'_k$ is a continuous quantity whereas $\mathcal{F}$ is defined on at discrete 2D locations. Thus, we use the differentiable bilinear sampling operation to sample from the discrete grid [25] to obtain the feature at $X_w^k$.

Such an operation has the highly desirable property that features from pixels in multiple images that may correspond to the same 3D world point unproject to the same location in the 3D grid - trivially enforcing epipolar constraints. As a result, any further processing on these unprojected grids has easy access to corresponding features to make matching decisions foregoing the need for long range image connections for feature matching in image space. Also, by projecting discrete 3D points into 2D and bilinearly sampling from the feature map rather than analytically tracing rays in 3D, we implicitly handle the issue where the probability of a grid voxel being hit by a ray decreases with distance from the camera due to their projective nature. In our formulation, every voxel gets a "soft" feature assigned based on where it projects back in the image, making the feature grids $\mathcal{G}^f$ smooth and providing stable gradients. This geometric procedure of lifting features from 2D maps into 3D space is in contrast with recent learning based approaches [5, 51] which either reshape flattened feature maps into 3D grids for subsequent processing or inject pose into the system using fully connected layers. This procedure effectively saves the network from having to implicitly *learn* projective geometry and directly bakes this given fact into the system. In LSMs, we use this operation to unproject the feature maps $\{\mathcal{F}_i\}_{i=1}^n$ in image space produced by the feature encoder into feature grids $\{\mathcal{G}_i^f\}_{i=1}^n$ that lie in metric 3D space.

For single image prediction, LSMs cannot match features from multiple images to reason about where to place surfaces. Therefore, we append geometric features along the rays during the projection and unprojection operation to facilitate single view prediction. Specifically, we add the depth value and the ray direction at each sampling point.

**Recurrent Grid Fusion.** The 3D feature grids $\{\mathcal{G}_i^f\}_{i=1}^n$ encode information about individual input images and need to be fused to produce a single grid so that further stages may reason jointly over all the images. For example, a simple strategy to fuse them would be to just use a point-wise function - *e.g.* max or average. This approach poses an issue where the combination is too spatially local and early fuses all the information from the individual grids. Another extreme is concatenating all the feature grids before further processing. The complexity of this approach scales linearly with the number of inputs and poses issues while processing a variable number of images. Instead, we choose to processed the grids in a sequential manner using a recurrent neural network. Specifically, we use a 3D convolutional variant of the Gated Recurrent Unit (GRU) [24, 4, 5] which combines the grids

$\{\mathcal{G}_i^f\}_{i=1}^n$ using 3D convolutions (and non-linearities) into a single grid $\mathcal{G}^p$. Using convolutions helps us effectively exploit neighborhood information in 3D space for incrementally combining the grids while keeping the number of parameters low. Intuitively, this step can be thought of as mimicking incremental matching in MVS where the hidden state of the GRU stores a running belief about the matching scores by matching features in the observations it has seen. One issue that arises is that we now have to define an ordering on the input images, whereas the output should be independent of the image ordering. We tackle this issue by randomly permuting the image sequences during training while constraining the output to be the same. During inference, we empirically observe that the final output has very little variance with respect to ordering of the input image sequence.

**3D Grid Reasoning.**   Once the fused grid $\mathcal{G}^p$ is constructed, a classical multi-view stereo approach would directly evaluate the photo-consistency at the grid locations by comparing the appearance of the individual views and extract the surface at voxels where the images agree. We model this step with a 3D UNet that transforms the fused grid $\mathcal{G}^p$ into $\mathcal{G}^o$. The purpose of this network is to use shape cues present in $\mathcal{G}^p$ such as feature matches and silhouettes as well as build in shape priors like smoothness and symmetries and knowledge about object classes enabling it to produce complete shapes even when only partial information is visible. The UNet architecture yet again allows the system to use large enough receptive fields for doing multi-scale matching while also using lower level information directly when needed to produce its final estimate $\mathcal{G}^o$. In the case of full 3D supervision (Voxel LSM), this grid can be made to represent a per voxel occupancy map. $\mathcal{G}^o$ can also be seen as a feature grid containing the final representation of the 3D world our system produces from which views can be rendered using the projection operation described below.

**Differentiable Projection.**   Given a 3D feature grid $G$ and a camera $\mathcal{P}$, the projection operation produces a 2D feature map $\mathcal{O}$ by gathering information along viewing rays. The direct method would be to trace rays for every pixel and accumulate information from all the voxels on the ray's path. Such an implementation would require handling the fact that different rays can pass through different number of voxels on their way. For example, one can define a reduction function along the rays to aggregate information (*e.g.* max, mean) but this would fail to capture spatial relationships between the ray features. Instead, we choose to adopt a plane sweeping approach where we sample from locations on depth planes at equally spaced z-values $\{z_k\}_{k=1}^{N_z}$ along the ray.

Consider a 3D point $X_w$ that lies along the ray corresponding to a 2D point $p$ in the projected feature grid at depth $z_w$ - *i.e.* $p = K[R|t]X_w$ and $z(X_w) = z_w$. The corresponding feature $\mathcal{O}(p)$ is computed by sampling from the grid $\mathcal{G}$ at the (continuous) location $X_w$. This sampling can be done differentiably in 3D using trilinear interpolation. In practice, we use nearest neighbor interpolation in 3D for computational efficiency. Samples along each ray are concatenated in ascending z-order to produce the 2D map $\mathcal{O}$ where the features are stacked along the channel dimension. Rays in this feature grid can be trivially traversed by just following columns along the channel dimension allowing us to *learn* the function to pool along these rays by using 1x1 convolutions on these feature maps and progressively reducing the number of feature channels.

**Architecture Details.**   As mentioned above, we present two versions of LSMs - Voxel LSM (V-LSM) and Depth LSM (D-LSM). Given one or more images and cameras, Voxel LSM (V-LSM) produces a voxel occupancy grid whereas D-LSM produces a depth map per input view. Both systems share the same set of CNN architectures (UNet) for the image encoder, grid reasoning and the recurrent pooling steps. We use instance normalization for all our convolution operations and layer normalization for the 3D convolutional GRU. In V-LSM, the final grid $\mathcal{G}^o$ is transformed into a probabilistic voxel occupancy map $\mathcal{V} \in R^{v_h \times v_w \times v_d}$ by a 3D convolution followed by softmax operation. We use simple binary cross entropy loss between ground truth occupancy maps and $\mathcal{V}$. In D-LSM, $\mathcal{G}^o$ is first projected into 2D feature maps $\{\mathcal{O}_i\}_{i=1}^n$ which are then transformed into metric depth maps $\{d_i\}_{i=1}^n$ by 1x1 convolutions to learn the reduction function along rays followed by deconvolution layers to upsample the feature map back to the size of the input image. We use absolute $L_1$ error in depth to train D-LSM. We also add skip connections between early layers of the image encoder and the last deconvolution layers producing depth maps giving it access to high frequency information in the images.

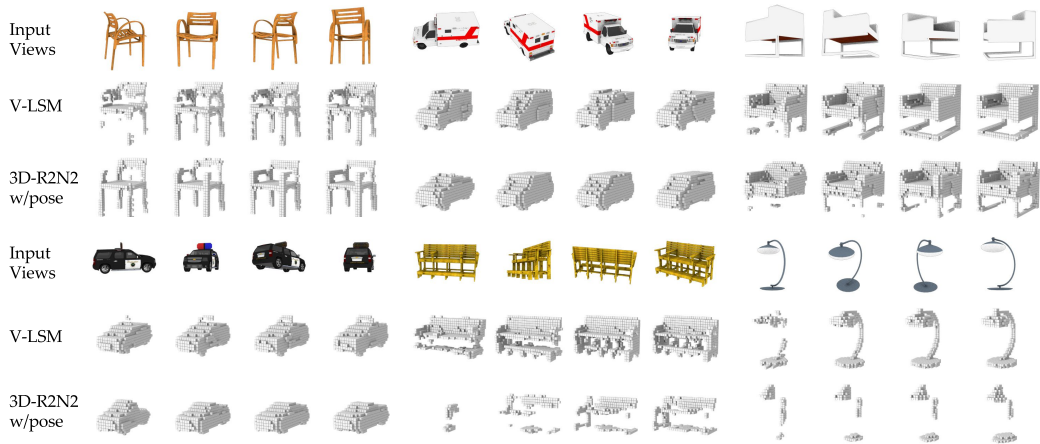

**Figure 3:** Voxel grids produced by V-LSM for example image sequences alongside a learning based baseline which uses pose information in a fully connected manner. V-LSM produces geometrically meaningful reconstructions (*e.g.* the curved arm rests instead of perpendicular ones (in R2N2) in the chair on the top left and the siren lights on top of the police car) instead of relying on purely semantic cues. More visualizations in supplementary material.

## 4 Experiments

In this section, we demonstrate the ability of LSMs to learn 3D shape reconstruction in a geometrically accurate manner. First, we present quantitative results for V-LSMs on the ShapeNet dataset [3] and compare it to various baselines, both classical and learning based. We then show that LSMs generalize to unseen object categories validating our hypothesis that LSMs go beyond object/class specific priors and use photo-consistency cues to perform category-agnostic reconstruction. Finally, we present qualitative and quantitative results from D-LSM and compare it to traditional multi-view stereo approaches.

**Dataset and Metrics.** We use the synthetic ShapeNet dataset [3] to generate posed image-sets, ground truth 3D occupancy maps and depth maps for all our experiments. More specifically, we use a subset of 13 major categories (same as [5]) containing around 44k 3D models resized to lie within the unit cube centered at the origin with a train/val/test split of $[0.7, 0.1, 0.2]$. We generated a large set of realistic renderings for the models sampled from a viewing sphere with $\theta_{az} \in [0, 360)$ and $\theta_{el} \in [-20, 30]$ degrees and random lighting variations. We also rendered the depth images corresponding to each rendered image. For the volumetric ground truth, we voxelize each of the models at a resolution of $32 \times 32 \times 32$. In order to evaluate the outputs of V-LSM, we binarize the probabilities at a fixed threshold (0.4 for all methods except visual hull (0.75)) and use the voxel intersection over union (IoU) as the similarity measure. To aggregate the per model IoU, we compute a per class average and take the mean as a per dataset measure. All our models are trained in a class agnostic manner.

**Implementation.** We use $224 \times 224$ images to train LSMs with a shape batch size of 4 and 4 views per shape. Our world grid is at a resolution of $32^3$. We implemented our networks in Tensorflow and trained both the variants of LSMs for 100k iterations using Adam. The projection and unprojection operations are trivially implemented on the GPU with batched matrix multiplications and bilinear/nearest sampling enabling inference at around 30 models/sec on a GTX 1080Ti. We unroll the GRU for upto 4 time steps while training and apply the trained models for arbitrary number of views at test time.

**Multi-view Reconstruction on ShapeNet.** We evaluate V-LSMs on the ShapeNet test set and compare it to the following baselines - a visual hull baseline which uses silhouettes to carve out volumes, 3D-R2N2 [5], a previously proposed system which doesn't use camera pose and performs multi-view reconstruction, 3D-R2N2 w/pose which is an extension of 3D-R2N2 where camera pose is injected using fully connected layers. For the experiments, we implemented the 3D-R2N2 system

| # Views | 1 | 2 | 3 | 4 |
|---|---|---|---|---|
| 3D-R2N2 [5] | 55.6 | 59.6 | 61.3 | 62.0 |
| Visual Hull | 18.0 | 36.9 | 47.0 | 52.4 |
| 3D-R2N2 w/pose | 55.1 | 59.4 | 61.2 | 62.1 |
| V-LSM | **61.5** | **72.1** | **76.2** | **78.2** |
| V-LSM w/bg | 60.5 | 69.8 | 73.7 | 75.6 |

**Table 1:** Mean Voxel IoU on the ShapeNet test set. Note that the original 3D-R2N2 system does not use camera pose whereas the 3D-R2N2 w/pose system is trained with pose information. V-LSM w/bg refers to voxel LSM trained and tested with random images as backgrounds instead of white backgrounds only.

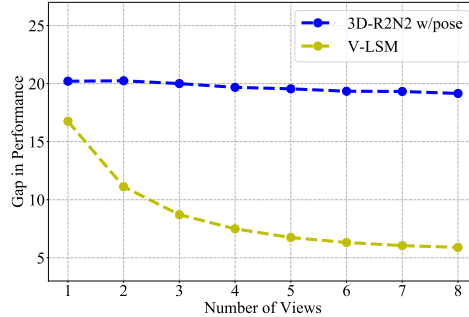

**Figure 4:** Generalization performance for V-LSM and 3D-R2N2 w/pose measured by gap in voxel IoU when tested on unseen object categories.

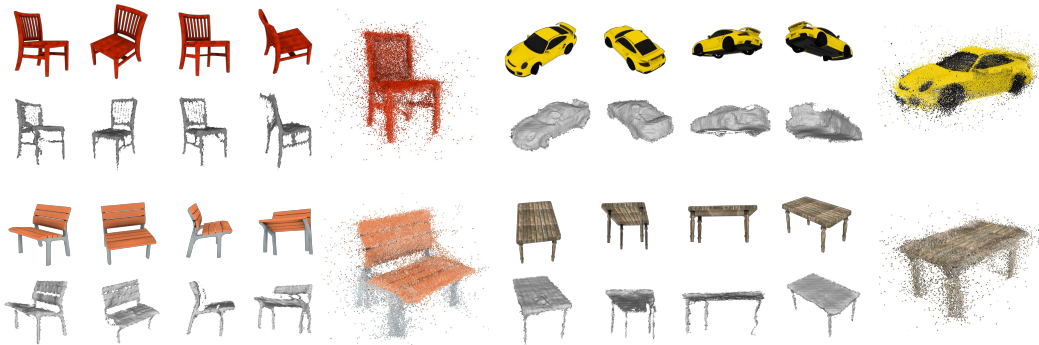

**Figure 5:** Qualitative results for per-view depth map prediction on ShapeNet. We show the depth maps predicted by Depth-LSM (visualized with shading from a shifted viewpoint) and the point cloud obtained by unprojecting them into world coordinates.

(and the 3D-R2N2 w/pose) and trained it on our generated data (images and voxel grids). Due to the difference in training data/splits and the implementation, the numbers are not directly comparable to the ones reported in [5] but we observe similar performance trends. For the 3D-R2N2 w/pose system, we use the camera pose quaternion as the pose representation and process it through 2 fully connected layers before concatenating it with the feature passed into the LSTM. Table 1 reports the mean voxel IoU (across 13 categories) for sequences of $\{1, 2, 3, 4\}$ views. The accuracy increases with number of views for all methods but it can be seen that the jump is much less for the R2N2 methods indicating that it already produces a good enough estimate at the beginning but fails to effectively use multiple views to improve its reconstruction significantly. The R2N2 system with naively integrated pose fails to improve over the base version, completely ignoring it in favor of just image-based information. On the other hand, our system, designed specifically to exploit these geometric multi-view cues improves significantly with more views. Figure 3 shows some example reconstructions for V-LSM and 3D-R2N2 w/pose. Our system progressively improves based on the viewpoint it receives while the R2N2 w/pose system makes very confident predictions early on (sometimes "retrieving" a completely different instance) and then stops improving as much. As we use a geometric approach, we end up memorizing less and reconstruct when possible. More detailed results can be found in the supplementary material.

**Generalization.** In order to test how well LSMs learn to generalize to unseen data, we split our data into 2 parts with disjoint sets of classes - split 1 has data from 6 classes while split 2 has data from the other 7. We train three V-LSMs - trained on split 1 (V-LSM-S1), on split 2 (V-LSM-S2) and both splits combined (V-LSM-All). The quantity we are interested in is the change in performance when we test the system on a category it hasn't seen during training. We use the difference in test IoU of a category $C$ between V-LSM-All and V-LSM-S1 if $C$ is not in split 1 and vice versa. Figure 4 shows the mean of this quantity across all classes as the number of views change. It can be seen that for a single view, the difference in performance is fairly high and as we see more views, the difference

in performance decreases - indicating that our system has learned to exploit category agnostic shape cues. On the other hand, the 3D-R2N2 w/pose system fails to generalize with more views. Note that the V-LSMs have been trained with a time horizon of 4 but are evaluated till upto 8 steps here.

**Sensitivity to noisy camera pose and masks.**
We conducted experiments to quantify the effects of noisy camera pose and segmentations on performance for V-LSMs. We evaluated models trained with perfect poses on data with perturbed camera extrinsics and observed that performance degrades (as expected) yet still remains better than the baseline (at $10°$ noise). We also trained new models with synthetically perturbed extrinsics and achieve significantly higher robustness to noisy poses while maintaining competitive performance (Figure 6). This is illustrated in Figure 6. The perturbation is introduced by generating a random rotation matrix which rotates the viewing axis by a max angular magnitude $\theta$ while still pointing at the object of interest.

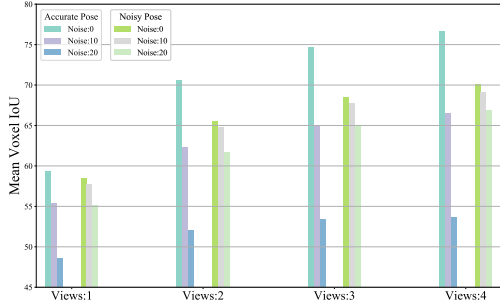

**Figure 6:** Sensitivity to noise in camera pose estimates for V-LSM for systems trained with and without pose perturbation.

We also trained LSMs on images with random images backgrounds (V-LSM w/bg in Table 1) rather than only white backgrounds and saw a very small drop in performance. This shows that our method learns to match features rather than relying heavily on perfect segmentations.

**Multi-view Depth Map Prediction.** We show qualitative results from Depth LSM in Figure 5. We manage to obtain thin structures in challenging examples (chairs/tables) while predicting consistent geometry for all the views. We note that the skip connections from the image to last layers for D-LSM do help in directly using low level image features while producing depth maps. The depth maps are viewed with shading in order to point out that we produce metrically accurate geometry. The unprojected point clouds also align well with each other showing the merits of jointly predicting the depth maps from a global volume rather than processing them independently.

**Comparision to Plane Sweeping.** We qualitatively compare D-LSM to the popular plane sweeping (PS) approach [6, 57] for stereo matching. Figure 7 shows the unprojected point clouds from per view depths maps produced using PS and D-LSM using 5 and 10 images. We omit an evaluation with less images as plane sweeping completely fails with fewer images. We use the publicly available implementation for the PS algorithm [19] and use 5x5 zero mean normalized cross correlation as matching windows with 300 depth planes. We can see that our approach is able to produce much cleaner point clouds with less input images. It is robust to texture-less areas where traditional stereo algorithms fail (*e.g.* the car windows) by using shape priors to reason about them. We also conducted a quantitative comparison using PS and D-LSM with 10 views (D-LSM was trained using only four images). The evaluation region is limited to a depth range of $\pm\sqrt{3}/2$ (maximally possible depth range) around the origin as the original models lie in a unit cube centered at the origin. Furthermore, pixels where PS is not able to provide a depth estimate are not taken into account. Note that all these choices disadvantage our method. We compute the per depth map error as the median absolute depth difference for the valid pixels, aggregate to a per category mean error and report the average of the per category means for PS as $0.051$ and D-LSM as $0.024$. Please refer to the supplementary material for detailed results.

## 5    Discussion

We have presented Learnt Stereo Machines (LSM) - an end-to-end learnt system that performs multi-view stereopsis. The key insight of our system is to use ideas from projective geometry to differentiably transfer features between 2D images and the 3D world and vice-versa. In our experiments we showed the benefits of our formulation over direct methods - we are able to generalize to new object categories and produce compelling reconstructions with fewer images than classical

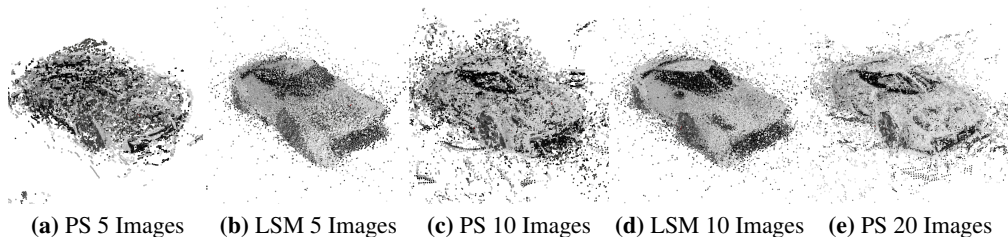

**(a)** PS 5 Images  **(b)** LSM 5 Images  **(c)** PS 10 Images  **(d)** LSM 10 Images  **(e)** PS 20 Images

**Figure 7:** Comparison between Depth-LSM and plane sweeping stereo (PS) with varying numbers of images.

systems. However, our system also has some limitations. We discuss some below and describe how they lead to future work.

A limiting factor in our current system is the coarse resolution ($32^3$) of the world grid. Classical algorithms typically work on much higher resolutions frequently employing special data structures such as octrees. We can borrow ideas from recent works [43, 21] which show that CNNs can predict such high resolution volumes. We also plan to apply LSMs to more general geometry than objects, eventually leading to a system which can reconstruct single/multiple objects and entire scenes. The main challenge in this setup is to find the right global grid representation. In scenes for example, a grid in terms of a per-view camera frustum might be more appropriate than a global aligned euclidean grid.

In our experiments we evaluated classical multi-view 3D reconstruction where the goal is to produce 3D geometry from images with known poses. However, our system is more general and the projection modules can be used wherever one needs to move between 2D image and 3D world frames. Instead of predicting just depth maps from our final world representation, one can also predict other view specific representations such as silhouettes or pixel wise part segmentation labels etc. We can also project the final world representation into views that we haven't observed as inputs (we would omit the skip connections from the image encoder to make the projection unconditional). This can be used to perform view synthesis grounded in 3D.

## Acknowledgments

This work was supported in part by NSF Award IIS- 1212798 and ONR MURI-N00014-10-1-0933. Christian Häne is supported by an "Early Postdoc.Mobility" fellowship No. 165245 from the Swiss National Science Foundation. The authors would like to thank David Fouhey, Saurabh Gupta and Shubham Tulsiani for valuable discussions and Fyusion Inc. for providing GPU hours for the work.

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
