[Supplementary Material]

# Learning a Multi-View Stereo Machine
# Supplementary Material

**Abhishek Kar**
UC Berkeley
akar@berkeley.edu

**Christian Häne**
UC Berkeley
chaene@berkeley.edu

**Jitendra Malik**
UC Berkeley
malik@berkeley.edu

## 1 Per-Category Results on ShapeNet

We present per category voxel IoU numbers for V-LSMs (Table 1) and 3D-R2N2 w/pose (Table 2) for all 13 classes in ShapeNet. We also present per category results for the quantitative comparison between D-LSM and plane sweep stereo in Table 3.

| Classes | aero | bench | cabinet | car | chair | display | lamp | speaker | rifle | sofa | table | phone | vessel | mean |
|---|---|---|---|---|---|---|---|---|---|---|---|---|---|---|
| **Views: 1** | 61.1 | 50.8 | 65.9 | 79.3 | 57.8 | 53.9 | 48.1 | 63.9 | 69.7 | 67.0 | 55.6 | 67.7 | 58.3 | 61.5 |
| **Views: 2** | 71.1 | 64.0 | 75.4 | 82.6 | 69.1 | 69.0 | 62.7 | 72.8 | 79.2 | 75.9 | 67.5 | 79.1 | 68.4 | 72.1 |
| **Views: 3** | 75.6 | 69.5 | 78.0 | 83.9 | 73.8 | 73.5 | 67.9 | 76.4 | 82.9 | 79.5 | 72.6 | 84.1 | 72.6 | 76.2 |
| **Views: 4** | 78.1 | 72.2 | 79.3 | 84.7 | 76.5 | 75.6 | 70.6 | 77.8 | 84.5 | 81.3 | 75.2 | 86.2 | 74.1 | 78.2 |

**Table 1:** Mean Voxel IoU for V-LSM for all classes in the ShapeNet test set.

| Classes | aero | bench | cabinet | car | chair | display | lamp | speaker | rifle | sofa | table | phone | vessel | mean |
|---|---|---|---|---|---|---|---|---|---|---|---|---|---|---|
| **Views: 1** | 56.7 | 43.2 | 61.8 | 77.6 | 50.9 | 44.0 | 40.0 | 56.7 | 56.5 | 58.9 | 51.6 | 65.6 | 53.1 | 55.1 |
| **Views: 2** | 59.9 | 49.7 | 67.0 | 79.5 | 55.0 | 49.8 | 43.1 | 61.6 | 59.9 | 63.9 | 56.0 | 70.4 | 57 | 59.4 |
| **Views: 3** | 61.3 | 51.9 | 69.0 | 80.2 | 56.8 | 53.3 | 44.2 | 62.9 | 61.0 | 65.3 | 58.0 | 73.4 | 58.9 | 61.2 |
| **Views: 4** | 62.0 | 53.0 | 69.7 | 80.6 | 57.7 | 55.1 | 44.5 | 63.5 | 61.6 | 66.3 | 58.8 | 74.3 | 59.5 | 62.1 |

**Table 2:** Mean Voxel IoU for 3D-R2N2 w/pose for all classes in the ShapeNet test set.

| Classes | aero | bench | cabinet | car | chair | display | lamp | speaker | rifle | sofa | table | phone | vessel | mean |
|---|---|---|---|---|---|---|---|---|---|---|---|---|---|---|
| **Plane Sweep** | 0.029 | 0.047 | 0.074 | 0.043 | 0.054 | 0.079 | 0.043 | 0.068 | 0.023 | 0.066 | 0.054 | 0.041 | 0.043 | 0.051 |
| **Depth LSM** | 0.024 | 0.030 | 0.022 | 0.016 | 0.023 | 0.028 | 0.027 | 0.029 | 0.023 | 0.021 | 0.025 | 0.021 | 0.027 | 0.024 |

**Table 3:** Mean depth map error ($L_1$ distance between predictions and ground truth at valid pixels) in the ShapeNet test set. Please refer to the main text for more details.

## 2 Output Visualization

We present additional qualitative results for V-LSMs and comparisons to the baseline system of 3D-R2N2 w/pose. Each box shows the input views in the top row, output from our proposed Learnt Stereo Machine (V-LSM) in the middle row and output from 3D R2N2 (with pose) in the last row.

**Figure 1:** Each box shows the input views in the top row, output from our proposed Learnt Stereo Machine (V-LSM) in the middle row and output from 3D R2N2 (with pose) in the last row.

**Figure 2:** Each box shows the input views in the top row, output from our proposed Learnt Stereo Machine (V-LSM) in the middle row and output from 3D R2N2 (with pose) in the last row.

**Figure 3:** Each box shows the input views in the top row, output from our proposed Learnt Stereo Machine (V-LSM) in the middle row and output from 3D R2N2 (with pose) in the last row.

**Figure 4:** Each box shows the input views in the top row, output from our proposed Learnt Stereo Machine (V-LSM) in the middle row and output from 3D R2N2 (with pose) in the last row.

**Figure 5:** Each box shows the input views in the top row, output from our proposed Learnt Stereo Machine (V-LSM) in the middle row and output from 3D R2N2 (with pose) in the last row.

**Figure 6:** Each box shows the input views in the top row, output from our proposed Learnt Stereo Machine (V-LSM) in the middle row and output from 3D R2N2 (with pose) in the last row.

**Figure 7:** Each box shows the input views in the top row, output from our proposed Learnt Stereo Machine (V-LSM) in the middle row and output from 3D R2N2 (with pose) in the last row.

**Figure 8:** Each box shows the input views in the top row, output from our proposed Learnt Stereo Machine (V-LSM) in the middle row and output from 3D R2N2 (with pose) in the last row.

**Figure 9:** Each box shows the input views in the top row, output from our proposed Learnt Stereo Machine (V-LSM) in the middle row and output from 3D R2N2 (with pose) in the last row.

**Figure 10:** Each box shows the input views in the top row, output from our proposed Learnt Stereo Machine (V-LSM) in the middle row and output from 3D R2N2 (with pose) in the last row.

**Figure 11:** Each box shows the input views in the top row, output from our proposed Learnt Stereo Machine (V-LSM) in the middle row and output from 3D R2N2 (with pose) in the last row.

**Figure 12:** Each box shows the input views in the top row, output from our proposed Learnt Stereo Machine (V-LSM) in the middle row and output from 3D R2N2 (with pose) in the last row.