[Reviews · NeurIPS 2017]

Reviewer 1



This submission proposes to learn 3D object reconstruction from a multi-view stereo setting end-to-end. Its main contribution is consistent feature projection and unprojection along viewing rays given camera poses and multiple images. Instead of mainly learning object shapes based on semantic image interpretation, this approach builds projective geometry into the approach to benefit from geometric cues, too. The paper proposes a way for differentiable unprojection that projects 2D image features to 3D space using camera projection equations and bilinear sampling. This is a simple yet effective way to implicitly build epipolar constraints into the system, because features from multiple views of the same world point unproject to the same voxel. While such combination of multiple views is of course not new, it is new as part of an end-to-end learnable approach. I did not completely understand how recurrent grid fusion works, please explain in more detail. A dedicated figure might ease understanding, too. I would also appreciate a short note on GPU RAM requirements and run-time. The paper is very well written and illustrations make it enjoyable to read. The approach is experimentally evaluated by comparison to traditional multi-view stereo matching algorithms (e.g., plane sweeping) and it is compared to the recent 3D-R2N2 approach, that also uses deep learning for 3D object reconstruction. While all experiments are convincing and support claims of the paper, it would be good to compare to more deep learning-based 3D reconstruction approaches (several are cited in the paper, e.g. [27] + additional work like "DeMoN: Depth and Motion Network for Learning Monocular Stereo" from Thomas Brox's lab). Clear explanation of theoretical differences to [27] would further improve the paper. At present, it remains unclear whether this submission is just a multi-view extension of [27] without reading [27] itself. Better motivating the own contribution here seems necessary. I believe this is a great paper and deserves to be published at NIPS, but also must admit that I still have some questions.

Reviewer 2



The paper presents an architecture for multi-view stereopsis. An image encoder (UNet) first processes the images to produce 2D feature maps which are unprojected into 3D world occupancy grids. These grids are fused sequentially using a RNN. Finally the resulting fused grid is processed with a 3D UNet architecture to produce the final grid. Feature projection and unprojection are formulated in a differentiable manner that allows training the architecture end-to-end. The proposed system can reconstruct voxel occupancy grids or depth maps of an object from one or several views. The approach is evaluated on the synthetic ShapeNet dataset. Strengths: - 3D reconstruction from multiple views is a fundamental problem in computer vision. - The paper is clear and reads well. - The methods outperforms previous work when the camera pose is known and the object is segmented. Weaknesses: - The applicability of the methods to real world problems is rather limited as strong assumptions are made about the availability of camera parameters (extrinsics and intrinsics are known) and object segmentation. - The numerical evaluation is not fully convincing as the method is only evaluated on synthetic data. The comparison with [5] is not completely fair as [5] is designed for a more complex problem, i.e., no knowledge of the camera pose parameters. - Some explanations are a little vague. For example, the last paragraph of Section 3 (lines 207-210) on the single image case. Questions/comments: - In the Recurrent Grid Fusion, have you tried ordering the views sequentially with respect to the camera viewing sphere? - The main weakness to me is the numerical evaluation. I understand that the hypothesis of clean segmentation of the object and known camera pose limit the evaluation to purely synthetic settings. However, it would be interesting to see how the architecture performs when the camera pose is not perfect and/or when the segmentation is noisy. Per category results could also be useful. - Many typos (e.g., lines 14, 102, 161, 239 ), please run a spell-check.

Reviewer 3



The paper presents a novel way of how to train a multi-view stereo system. I think this ieda is valuable, and the paper is well presented and thouroughly written. Some minor remarks: - it is not clear from the beginning if the cameras are calibrated or if the relative camera poses are known for the system. - In the evaluation (table 1), visual hull is taken as baseline. I think this is rather unfair, because visual hull only uses silhouettes as input and not full images. - Better indeed was to compare with classical existing stereo mehtods, such as plane sweeping. Why is this method only qualitatively compared to the proposed method (fig. 6)? Why is no quantitively comparison made? - The resulting point clouds still show a large amount of noise voxels flowing around the generated 3D model. Can't you use a noise removal technique, just to create a visually more attractive result?